# Parents’ Perception of Food Insecurity and of Its Effects on Their Children in Italy Six Months after the COVID-19 Pandemic Outbreak

**DOI:** 10.3390/nu13010121

**Published:** 2020-12-31

**Authors:** Arianna Dondi, Egidio Candela, Francesca Morigi, Jacopo Lenzi, Luca Pierantoni, Marcello Lanari

**Affiliations:** 1Pediatric Emergency Unit, Scientific Institute for Research and Healthcare (IRCCS), Sant’Orsola University Hospital, 40138 Bologna, Italy; egidiocandela@gmail.com (E.C.); francesca.morigi@gmail.com (F.M.); luca.pierantoni@aosp.bo.it (L.P.); marcello.lanari@unibo.it (M.L.); 2Department of Biomedical and Neuromotor Sciences, Alma Mater Studiorum—University of Bologna, 40126 Bologna, Italy; jacopo.lenzi2@unibo.it

**Keywords:** SARS-CoV-2 pandemic, COVID-19, lockdown, food insecurity, junk food, body weight, pediatric, children

## Abstract

The Severe Acute Respiratory Syndrome Coronavirus-2 (SARS-CoV-2) pandemic and subsequent containment measures are causing an increase in food insecurity (FI) worldwide, with direct consequences on children’s nutrition. We aimed to investigate the effects of the lockdown imposed in Italy on FI and changes in eating habits and body weight in the pediatric population 6 months after the beginning of the pandemic through a cross-sectional online survey proposed to parents of children <18 and living in Italy. Among 5811 respondents, most of whom were Italian, living in Northern Italy, and with a wealthy household economy, 8.3% reported that their families were at risk of FI before the appearance of SARS-CoV-2 and, alarmingly, this percentage increased to 16.2% after the pandemic began, with households from Southern Italy being more at risk. Moreover, 27.3% of the parents reported that their children were eating more; an increase in “junk food” consumption was also found; 31.8% of the respondents declared an increase in children’s weight; weight loss prevailed among adolescents. Since the SARS-CoV-2 pandemic is again requiring restrictions, our findings might serve as a warning to politicians to promote healthy lifestyles and provide assistance to the groups in need.

## 1. Introduction

Food insecurity (FI), which refers to a lack of access by all people at all times to enough food for an active and healthy life [1], is a striking issue impacting especially the health of young individuals and their development through inadequate nutrition in terms of hunger, undernutrition, overnutrition with low-quality, sugary and fatty food, and an insufficient intake of macronutrients and micronutrients, often referred to as “hidden hunger” to describe the invisible nature of the problem and the lack of overt symptoms [2,3,4]. The prevalence of FI has increased during recent years in several European countries [5,6,7]. This phenomenon has been noticed also in Italy, where FI increased from 7% in 2009 to 17% in 2012 [8].

The Severe Acute Respiratory Syndrome Coronavirus-2 (SARS-CoV-2) pandemic and all the containment measures implemented to slow its spread [9] have caused one of the most serious crises in the world economy since the end of the Second World War [10], and most countries are now beginning to deal with the dramatic societal and economic consequences of it. In fact, the pandemic is affecting food systems both directly, through impacting on food availability and demand, and indirectly, through decreasing the capacity to produce and distribute food [11]. People suffering from job loss or disruption have been reported to be at higher risk of household FI since the SARS-CoV-2 pandemic [12]. Furthermore, with SARS-CoV-2-related school closures occurring in nearly 200 countries across the world, including Italy where schools were closed from February 24 2020, and re-opened only in September 2020, children have been missing out on school meals [13], impacting not only on children’s nutrition and lifestyle, but also on the most vulnerable families by reducing their income and exacerbating FI.

Higher FI may also act as a multiplier for the epidemic due to its negative health effect [10]. Some authors state that inadequate nutrition can contribute to the spread of pandemic disease [14], since it decreases immune function [15]. Moreover, data suggest that hunger reduces the ability to resist to a viral infection and, hence, makes diseases more severe [16]. In confirmation of this, malnutrition and obesity have been related to increasing rates of Coronavirus Disease (COVID)-19 hospitalization in the United States [17]. On the other hand, fear and uncertainty correlated to the disease may lead to overeating, as stress is known to increase food consumption [18].

Furthermore, quarantine can cause a high prevalence of psychological distress [19]. It has been reported that quarantined people are significantly more likely to report exhaustion [20], depression [21], stress [22], low mood, irritability, insomnia [23], and post-traumatic stress symptoms in both parents and children [24,25].

Following these considerations, we hypothesized that social distancing measures to fight the COVID-19 pandemic would strongly impact on a vulnerable population such as children in terms of FI and related inadequate nutrition issues. To investigate this hypothesis, we inquired about Italian parents’ perception of the effects of the lockdown imposed for SARS-CoV-2 infection containment on FI, eating habits, and weight changes on their children 6 months after the beginning of the pandemic.

## 2. Materials and Methods

### 2.1. Study Design and Survey Development

A cross-sectional online survey of 78 questions was proposed to a population made of parents with children up to 18 years old and living in Italy. The questions were elaborated on the basis of a review of the scientific literature relating to the topic and extensive online discussion between the authors.

The questionnaire, which is described in detail in Appendix A, investigated the demographic variables of families and some social determinants of health (housing, level of parental education, employment, hunger, stress) [26]. For the present study, we examined in particular the questions about FI, which was investigated through the Hunger Vital Sign [27] adapted to evaluate FI before and after the beginning of the pandemic, and those about changes in children’s eating habits, particularly in the intake of “junk foods” such as snacks, sweetened fruit juices and soft drinks, and in weight variation. We measured the degree of interrelation of these questions with the family composition, parental education, parents’ and children’s mood changes.

### 2.2. Survey Conduction and Distribution

The online survey was conducted on the research platform managed by Qualtrics. The Qualtrics model allows researchers to develop surveys using Qualtrics software and is able to produce online reliable data similarly to telephone and in-person traditional methods [28].

The survey was open from 1 September until 15 October, 2020, but most of the responses (93.3%) were recorded between 20 September and 4 October, 2020. It was distributed through a link and a QR code disseminated through posters affixed in a tertiary care pediatric hospital and in the waiting rooms of the pediatricians’ offices, and through social networks with a snowball sampling technique. The questionnaire was filled out anonymously by parents, spontaneously and with prior online informed consent. The participants were given no reward or incentive.

### 2.3. Ethics

The present study was approved by the Ethics Committee of the University Hospital of Bologna (Italy) (Institutional Review Board approval number 762/2020/Oss/AOUBo).

### 2.4. Statistical Analysis

Summary statistics were presented as frequencies and percentages. Differences in food insecurity before and after the pandemic were investigated with the Wilcoxon signed-rank test. Multivariable logistic regression analysis was used to investigate which sociodemographic characteristics were associated with increased food insecurity and changes in eating habits after the pandemic. Covariates that were not significantly associated with the outcome with a significance level of *p* < 0.25 at bivariate analyses were not included in multivariable logistic regression.

Because one covariate (smart working) had a high proportion of missing data (≈25%), multiple imputation by chained equations was used to replace missing values with multiple sets of simulated values to complete the data (*m* − 20). Regression estimates from the multiply imputed sets were then combined into one overall estimate with an associated variance that incorporated the within- and between-imputation variability [29]. Complete-case analysis of the other data was used, with the exception of children’s disorders and disabilities (missingness > 1%), for which missing information was treated as a distinct category. Distance learning was excluded from multivariable analysis due to its strong correlation with children’s age; no other collinearity issues were found.

All data were analyzed using Stata version 15 (StataCorp. 2017. Stata Statistical Software: Release 15. College Station, Tx: StataCorp LP). To control for type I error related to multiple testing, the significance level was set at 0.01. Odds ratios from multivariable logistic regression were tested using the 2-sided Wald test.

## 3. Results

Overall, 7958 people filled in the questionnaire, but after exclusion of the incomplete answers about FI and eating habits, 6094 responses (76.6%) were included in the present study. A further 283 subjects (3.6%) were excluded due to the presence of missing data on the variables of interest (see Figure A1 in Appendix B).

The socioeconomic characteristics of the study sample are summarized in Table 1, while experience and attitudes toward the COVID-19 pandemic are summarized in Table 2. The majority of the respondents were female (91.7%), Italians (97.1%) and living in Northern Italy (89.0%). Most of them had a high school or university grade (96.0%), in 85.8% of the families both parents were employed, and more than 90% described their household economy as either “well-off” or “overall satisfactory”. Around 40% of the respondents reported that, since the beginning of the pandemic, their household economy somehow worsened, and more than 50% considered their means more difficult or at risk; in 4.4% of the cases either parent lost their job since the beginning of the pandemic. Concerning attitudes during the lockdown, parents thought that their children had mainly missed meeting friends and reported that they had become more nervous and had feelings of loneliness in 73.5% and 68.9% of cases, respectively.

FI before and after the beginning of the pandemic is illustrated in Figure 1 and Figure 2. We found that the frequency with which parents worried about their food supplies significantly changed after the outbreak (Wilcoxon signed-rank *z*-test—19.91, *p* <0.001). More specifically, concern got worse in 616 parents (10.6%) and got better in 88 (1.5%). Similarly, 144 families (2.5%) ran out of food more often than before the pandemic, and 56 (1.0%) less often (*z*-test—6.23, *p* < 0.001).

Table 3 reports the multivariable logistic regression analysis for FI since the pandemic outbreak. Families from Southern Italy, with a household difficult or unsustainable economic situation, with more than one child, with at least one parent on furlough, with any worsening of the household economy, or in which the parent considered their means more at risk, showed a significantly higher risk of becoming food insecure. Parents’ aged over 50 years, higher parents’ school grade, and both parents being employed appeared as protective factors against an increase in FI. Moreover, a significant association was documented between households running out of food and children being more nervous, accounting for the impact of FI on children’s mood.

Changes in children’s food intake and body weight are illustrated in Figure 3. Globally, children’s food intake changed in 40.2% of the sample. Among the 1588 families that declared that their children were eating more food (27.3% of the total sample), there was an increase in the consumption of snacks in 958 (60.3%), fruit juices in 223 (14.0%), and soft drinks in 165 (10.4%). Table 4 reports the results of the logistic regression analysis for food intake. Factors associated with increased children’s food intake were parental female sex, difficult or unsustainable household economy, having more than one child, having kids older than 2, and children missing outdoor activities. Interestingly, all mood changes (parent’s perception of more difficult or at-risk familial means, children being more nervous or with an improvement in their mood, children feeling lonely) were associated with a modification in children’s food consumption, either increased or lowered.

Multivariable logistic regression analysis for body weight is reported in Table 5. Having more than one child or having children aged 6 to 10 years were factors associated with children’s weight gain. Any difficulty in the household economy, having children in their adolescence, and having children with learning disabilities were associated with children’s weight loss. Mood changes (children having feelings of loneliness or missing going to school) were associated with both weight gain and weight loss.

## 4. Discussion

A severe increase in FI after the SARS-CoV-2 pandemic is highly feared and is expected to affect mostly vulnerable groups like children [30,31]. Early evidence suggests that FI is indeed rapidly rising [32,33]. According to the first projections of the World Food Program and United Nations Food and Agriculture Organization (FAO), the SARS-CoV-2 pandemic could almost double the number of people suffering from FI [34]. Among other studies, Niles et al. reported a 32.3% increase in FI in the Vermont population since the COVID-19 outbreak, and the odds of experiencing FI were higher among households with children [12].

Our questionnaire was proposed to all families living in Italy and with at least one child younger than 18. Most of the respondents (89%) were from Northern Italy, mainly from those regions (Emilia-Romagna, Lombardy, and Veneto) where the first wave of SARS-CoV-2 infection was earlier and particularly severe. This area is also recognized as one of the richest and wealthiest in Europe [35] and its health system has always been considered strong and efficient [36].

Overall, 8.4% of the respondents reported that their families were at risk of FI before the appearance of SARS-CoV-2. Alarmingly, even if most of the families lived in this wealthy area, this percentage increased to 16.5% 6 months after the beginning of the pandemic. Moreover, there was a significant increase both in the households who worried about running out of food and in those that actually did. The latter are likely people who were already subject to an unstable household economy, and in whom the containment measures for the pandemic caused a dramatic worsening of their already precarious condition.

We found that parents’ higher age (over 50), higher parents’ school grade, and both parents being employed were protective against FI. Similar protective factors had already been identified in several studies before the pandemic [37,38]. Recently, an Australian online survey, performed between May and June 2020 on 1170 adults living in Tasmania, found that increasing age, a university education and higher income were protective against FI [39].

The risk factors that we identified for an increase in FI are not surprising, as they are all connected with lower disposable economic means: among them, it is noteworthy that families living in Southern Italy, which is less wealthy than Northern Italy, had a higher risk of FI [40]. The risk for FI drops quickly with income [41], and several authors report that also households with children, higher size households, a single, divorced or separated parent, especially if a woman, or an unmarried parent in a more complex family (for example, one that includes a cohabiting partner or another adult such as a grandparent) are factors more frequently associated with FI [42,43,44,45]. Furthermore, a renter, a younger, or a less educated person is more likely to be food insecure than their respective counterparts [46]. Immigrants are well-known to be at a higher risk of FI [47] and are usually one of the groups where the risk factors for FI that we identified are predominant (household difficult or unsustainable economic situation, more than one child, any worsening of the household economy, families in which the parent considered their means more at risk). Unfortunately, this population subgroup was only included in the present survey in a limited way, and the 3% respondents who were not of Italian origin did not appear to suffer from FI more than those of Italian origin. Interestingly, a study by Morales et al. highlighted that, among 74413 US households, Blacks, Asians, Hispanics, or other racial/ethnic minorities were not significantly more food insecure than White households during the pandemic [48]. As a matter of fact, global COVID-19 data reveal a concrete and urgent threat to food security [49,50]. The number of people suffering from chronic hunger can rise dramatically, and the situation may worsen for those who already suffer from FI [51]. The reduction of access to food and other essential items is expected to magnify disparities in healthy lifestyle behaviors and increase social tensions, migration, and severe malnutrition, which has been associated with a higher risk for more severe cases of SARS-CoV-2 infection [52]. International agencies point out the necessity to adopt measures to facilitate the flow of food, support the most vulnerable, ensure access to adequate and healthy food, and invest in sustainable and resilient food systems [53].

A worrisome effect of the containment measures to slow SARS-CoV-2 spread is the increase in food intake and consumption of the so-called “junk food”. In our population, 27.5% of the parents declared that their children were eating more since the beginning of the pandemic, and this happened more frequently in those families in which parents worried about running out of food (i.e., households at risk of FI). Moreover, an increase in snacks, sugared fruit juices, and soft drinks consumption was reported. Another online survey, performed in the USA during the months of April and May 2020 on 584 parents with children, found a 17% overall decrease of food-secure families and a 20% increase of families experiencing very low food security since the beginning of the pandemic; moreover, about one-third of the families reported an increase in the amount of high-calorie snack foods and desserts/sweets during the same period [54]. An Italian study including 41 children and adolescents with obesity found that the analyzed subjects consumed more potato chips, red meat, sugary drinks, but also fruit, after 3 weeks of mandatory stay-at-home confinement during the pandemic [55]. Interestingly, we found a correlation between children missing outdoor activities and major food intake. Evidence suggests that greater time spent outside is associated with healthier dietary patterns (i.e., reduced snacking, more inhibited eating behavior) [56]. We assume that children that were used to spend more time outside changed their dietary pattern, and this resulted in an increase in food intake.

An increase in children’s body weight was declared by 32% of the respondents. Other authors reported similar changes both in pediatric [57] and in adult [58,59,60,61] populations since the beginning of the restrictions due to the pandemic. A study analyzing the Google Trends covering a timeframe from June 2019 to April 2020 revealed that people have replaced outdoor activities with sedentary indoor behaviors [62]. The Italian lockdown strict rules did not allow children to do outdoor activities and go to school, and an increase in screen time was reported [55], making this period unfavorable for maintaining healthy lifestyle behaviors. Furthermore, weight gain and pediatric obesity are strictly linked to FI among low-income groups [47,63], but it is not clear whether FI is associated with obesity above and beyond the influence of poverty. In fact, in our study weight gain was not significantly higher in those households at risk of FI. Indeed, poverty is strictly related to weight gain, given the availability of high energy, sugary and fatty products, and beverages that are easily purchased at low prices [47].

Furthermore, in our study the feeling of loneliness was associated with both weight gain and weight loss. Delgado Floody et al. had already reported an association between sadness or loneliness and obesity in children [64]. In fact, in children, stress can stimulate eating in the absence of hunger, which could facilitate overweight [65]. A Spanish study highlighted large weight changes including weight loss during the COVID19 home confinement in 4379 adolescents and adults, speculating that COVID19 stressors could have triggered depressive symptoms leading to maladaptive food behaviors (overeating, restrictive eating) [66].

Interestingly, we found that children older than 14 years were significantly at risk of weight loss during lockdown. This enlightens the risk of weight loss in adolescence during a tough period, as we know that weight and shape concern correlated substantially with emotional and stressful problems in the teenage years [67]. Gallè et al. also reported that, among undergraduate students, 41% ate less or better during the pandemic than before, while only 17.1% declared to eat more [68]. Among teenagers, evidence supports a strong association between weight/shape concern and low self-esteem, as well as emotional problems, including depressive symptoms and anxiety [67]. In a school-based survey on 4746 adolescents, the relationship between body dissatisfaction and self-esteem was strong and significant in both sexes, and did not differ significantly between middle school and high school cohorts, indicating that body dissatisfaction and self-esteem are strongly related among nearly all groups of adolescents [69]. Both low self-esteem and body dissatisfaction early in life have been found to predict adverse health outcomes later in life, including the use of unhealthy weight-control behaviors, other eating-disordered behaviors, general psychological distress, and a variety of other negative outcomes [70,71,72]. In stressful times like those of an active pandemic with severe containment measures, monitoring such possible effects among teenagers should not be forgotten.

As learned from the Severe Acute Respiratory Syndrome (SARS) epidemics in 2002–2004, quarantine, especially if mandatory, can result in a high prevalence of psychological distress, manifested by irritability, exhaustion [19], hyperphagia and binge-eating, both eventually resulting in a weight change [73]. Prolonged staying at home may also support eating palatable meals and snacking, affect individual choices to cook more or buy prepared food more often, and it is often related to lower physical activity. A Polish study identified a high frequency of dietary habit modification, manifested by eating (43.5% of respondents) and snacking (51.8%) more, resulting in a weight increase in 29.9% of adult individuals [59]. Variations in mental health may contribute to or impair healthy eating, in a bi-directional relation between eating and mental health: one’s mood or psychological state can affect what and how much one eats and eating affects one’s mood and psychological well-being [74]. A recent study revealed the importance of mood as the leading food choice motive in 48% of 938 adults interviewed about their food consumption during the COVID-19 lockdown in France [75]. This is in line with the results of our study: all mood changes in both parents and children were associated with a modification in children’s food consumption, either higher or lower; mood changes in children were mainly associated with weight gain, but also with weight loss.

Limitations of this study include, first, that this survey was promoted predominantly by social media, therefore we cannot know the real number of people who received the invitation to participate to the survey and, consequently, the response rate. Moreover, this design of the study could limit the ability to reach groups of the population that do not have access to the Internet and might be at higher risk of FI and poverty. Of note, few respondents to the present survey were immigrants and with a difficult or unsustainable household economy; however, our results underline a significant worsening in FI even in a population apparently less at risk. Thirdly, we did not collect anthropometric data such as weight and height, which would allow calculation of the body mass index and its change in the analyzed period. Finally, children’s mood and feelings were evaluated through parents’ perception and not directly through children’s perception. Similarly, the questionnaire did not measure the actual food consumption, but the perceived changes in food intake, nor the actual economic status, but its perception.

Despite these limitations, this study seems to confirm our initial hypothesis that social distancing measures to fight COVID-19 pandemic may be impacting on vulnerable populations such as children and their families in terms of worsening in FI and related inadequate nutrition issues with consequent weight changes which can be worrisome in specific age groups.

## 5. Conclusions

In conclusion, our study suggests that children and their families may be already suffering from an increase in FI even in the wealthier Italian areas, and that households living in Southern Italy and with lower income and disposable means are especially at risk; health policies should keep a close eye on these groups. According to Leddy and colleagues, in the short-term, increased FI may contribute to acute chronic disease complications, and in the long term to increased risk for chronic disease development, morbidity, and mortality [76]. It is essential that health care and social protection systems take into account principles of equity and sustainability to ensure food and nutrition security for all. Scientific research should work to fill in the major knowledge gaps about food and nutrition implications of the pandemics and effective delivery of equitable social protection programs and policies in these circumstances [31]. Political restriction measures should consider the potential negative effects that they can induce on nutrition and health of children and their families. Since the second wave of the SARS-CoV-2 pandemic is again challenging the world’s health systems and economies and is again requiring social and movement restrictions to control the spread of the virus, our findings might serve as a warning to politicians to take ad hoc measures for fragile groups such as children.

## Figures and Tables

**Figure 1 nutrients-13-00121-f001:**
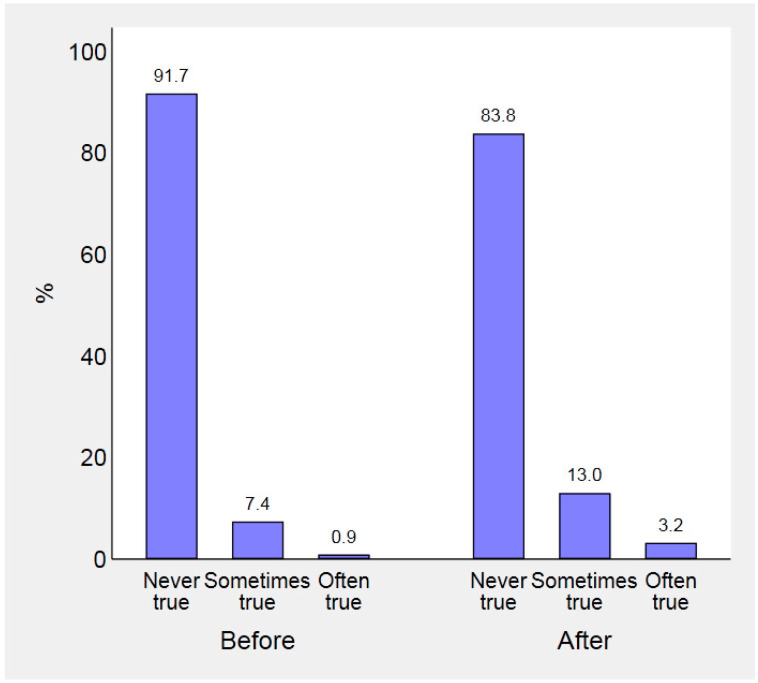
Frequency with which parents worried about running out of food, in the 12 months preceding and in the 6 months following COVID-19 outbreak.

**Figure 2 nutrients-13-00121-f002:**
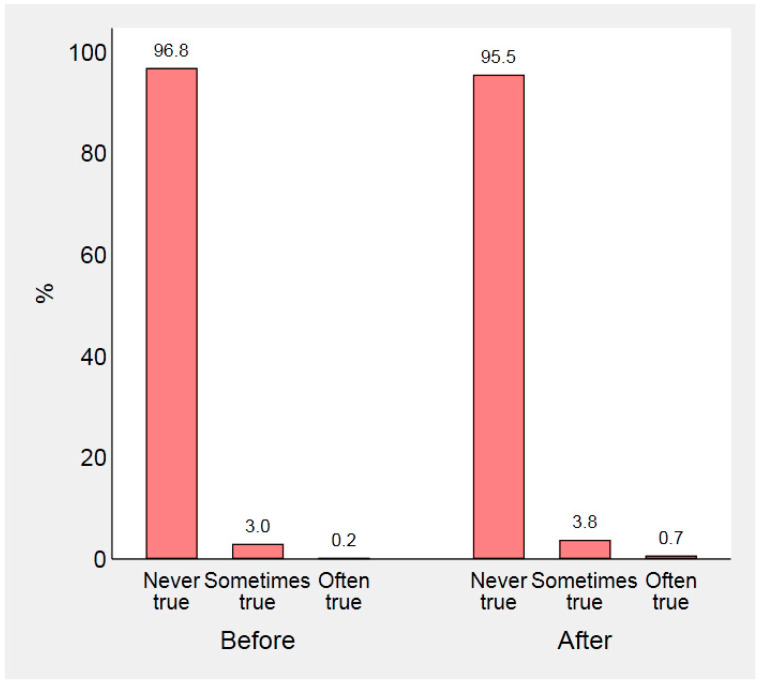
Frequency with which parents ran out of food and did not have enough money to buy more, before and after COVID-19 outbreak.

**Figure 3 nutrients-13-00121-f003:**
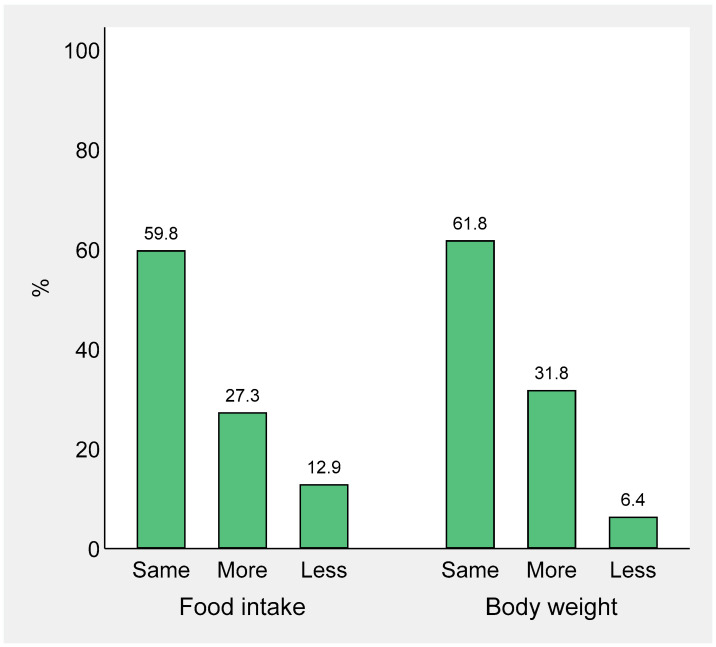
Changes in children’s food intake and body weight after COVID-19 outbreak.

**Table 1 nutrients-13-00121-t001:** Sociodemographic characteristics of the study sample (*n* − 5811).

Characteristic	*n*	%
*Age, y*		
≤30	246	4.2
31–35	910	15.7
36–40	1696	29.2
41–45	1587	27.3
46–50	943	16.2
>50	429	7.4
*Sex*		
Female	5327	91.7
Male	484	8.3
*Country of origin*		
Italy	5645	97.1
Outside Italy *	166	2.9
*Area of residence* †		
Northern Italy	5171	89.0
Central Italy	348	6.0
Southern Italy	292	5.0
*Educational attainment*		
Middle school (8th grade) or less	230	4.0
High school (12/13th grade)	2100	36.1
University degree or more	3481	59.9
*Educational attainment of the other parent*		
Middle school (8th grade) or less	889	15.3
High school (12/13th grade)	2588	44.5
University degree or more	2334	41.2
*Working condition of the parents*		
Both employed	4983	85.8
One unemployed	794	13.7
Both unemployed	34	0.6
*Job type*		
Clerk	2913	50.1
Retired	781	13.4
Homemaker	648	11.2
Freelancer	276	4.7
Laborer	263	4.5
Health-care worker	132	2.3
Dealer	118	2.0
Other ‡	680	11.7
Economic status		
Well-off	2307	39.7
Somewhat difficult but overall satisfactory	3123	53.7
Quite difficult	356	6.1
Often unsustainable	25	0.4
*Number of children in the family*		
1	2405	41.4
2	2827	48.6
3	446	7.7
>3	133	2.3
*Age of the youngest or only child, y*		
≤2	1759	30.3
3–5	1419	24.4
6–10	1523	26.2
11–14	685	11.8
>14	425	7.3
*Children with disorders or disabilities*		
No	5147	88.6
Learning disabilities	252	4.3
Other disabilities	85	1.5
Chronic conditions	67	1.2
Autism spectrum disorders	48	0.8
Multiple conditions	47	0.8
Unspecified	165	2.8
*The parents live together*		
Yes	5329	91.7
No	431	7.4
Single parenthood	51	0.9
*Other relatives at home*		
No	5397	92.9
Grandparents	329	5.7
Other relatives	85	1.5
*Type of accommodation*		
Own property	5083	87.5
Renting	720	12.4
Hotel/pension, reception center or with others	8	0.1

* 120 from Europe, 28 from the Americas, 11 from Africa and 7 from Asia. † Northern Italy: Piedmont, Aosta Valley, Lombardy, Liguria, Trentino-South Tyrol, Veneto, Friuli-Venezia Giulia, Emilia-Romagna; Central Italy: Tuscany, Umbria, Marche, Lazio; Southern Italy: Abruzzo, Molise, Campania, Apulia, Basilicata, Calabria, Sicily, Sardinia. ‡ Including teachers, students, educators, farmers, police, unemployed, and unable to work.

**Table 2 nutrients-13-00121-t002:** Parents’ experience and attitudes toward the coronavirus pandemic (*n* − 5811).

Characteristic	*n*	%
*A member of the family got COVID-19*		
No	5031	86.6
Yes	420	7.2
Yes, hospitalized	192	3.3
Yes, passed away	168	2.9
*Smart working for either parent*		
Yes	2925	50.3
No	1464	25.2
Unspecified	1422	24.5
*Furlough for either parent*		
No	3587	61.7
Yes	2224	38.3
*Economic status after the outbreak*		
Improved	159	2.7
Unchanged	3203	55.1
Slightly worsened	2161	37.2
Worsened	246	4.2
Become critical	42	0.7
*Either parent has lost their job*		
No	5558	95.6
Yes	253	4.4
*How the parent sees her/his means after the pandemic*		
Better	126	2.2
Unchanged	2349	40.4
More difficult	2938	50.6
Much more difficult	332	5.7
Seriously at risk	66	1.1
*What have your children missed more? **		
Going to school	1770	30.5
Outdoor activities	2723	46.9
Meeting friends	4676	80.5
Meeting relatives	2411	41.5
Playing sports	2462	42.4
*Any mood swing in your children?*		
Yes, more nervous, troubled or sad	4269	73.5
No	1389	23.9
Yes, their mood has improved	153	2.6
*Did your children have feelings of loneliness?*		
Yes, putting it into words	2151	37.0
Yes, not putting it into words	1852	31.9
No	1808	31.1
*Any child attending distance-learning classes?*		
Yes	3737	64.3
No	2074	35.7

* Multi-select question (the sum of percentages exceeds 100).

**Table 3 nutrients-13-00121-t003:** Multivariable logistic regression analysis of increased vs. unchanged/decreased food insecurity after COVID-19 outbreak. Results are presented as odds ratios (ORs) and 99% confidence intervals (CIs).

Characteristic	Worrying About Food Supplies	Running Out of Food
	OR (99% CI)	OR (99% CI)
*Age, y (ref: ≤30)*		
31–35	1.16 (0.66–2.04)	0.66 (0.26–1.71)
36–40	1.03 (0.59–1.78)	1.08 (0.45–2.62)
41–45	0.77 (0.44–1.35)	0.61 (0.24–1.56)
46-50	0.68 (0.37–1.25)	0.54 (0.19–1.53)
>50	0.37 (0.17–0.81) *	0.56 (0.16–2.00)
*Sex (ref: male)*		
Female	1.38 (0.81–2.35)	1.38 (0.46–4.16)
*Country of origin (ref: Italy)*		
Outside Italy	1.28 (0.70–2.32)	1.34 (0.51–3.50)
*Area of residence (ref: Northern Italy)*		
Central Italy	0.89 (0.54–1.48)	0.88 (0.34–2.23)
Southern Italy	1.66 (1.01–2.72) *	1.56 (0.59–4.11)
*Educational attainment of the parents (ref: both secondary school)*		
Secondary school & graduate school	0.68 (0.50–0.92) *	0.68 (0.38–1.20)
Both graduate school	0.69 (0.48–0.98) *	0.43 (0.18–1.02)
*Working condition of the parents (ref: either unemployed)*		
Both employed	0.63 (0.42–0.94) *	1.00 (0.48–2.10)
*Job type (ref: health-care worker)*		
Clerk	1.40 (0.56–3.54)	0.97 (0.16–5.82)
Retired	1.62 (0.60–4.38)	0.73 (0.10–5.34)
Homemaker	1.75 (0.66–4.64)	0.63 (0.09–4.32)
Freelancer	0.87 (0.34–2.25)	0.67 (0.11–4.08)
Laborer	1.52 (0.56–4.11)	1.50 (0.24–9.46)
Dealer	2.19 (0.71–6.77)	1.06 (0.13–8.91)
Other	1.40 (0.54–3.64)	1.03 (0.16–6.56)
*Economic status (ref: well-off)*		
Somewhat difficult but overall satisfactory	2.11 (1.48–3.01) *	5.18 (1.34–20.06) *
Difficult/unsustainable	3.90 (2.39–6.38) *	10.72 (2.53–45.33) *
*Number of children (ref: 1)*		
2	1.37 (1.05–1.79) *	0.81 (0.48–1.37)
3	1.43 (0.91–2.26)	1.15 (0.50–2.63)
>3	1.74 (0.83–3.67)	1.63 (0.42–6.37)
*Type of accommodation (ref: own property)*		
Renting/other	1.19 (0.86–1.63)	1.68 (0.99–2.86)
*Smart working for either parent (ref: no)*		
Yes	0.95 (0.69–1.32)	0.90 (0.47–1.70)
*Furlough for either parent (ref: no)*		
Yes	1.43 (1.10–1.87) *	1.21 (0.72–2.02)
*Economic status after the outbreak (ref: improved/unchanged)*		
Slightly worsened	2.14 (1.57–2.92) *	7.96 (2.85–22.21) *
Worsened	5.77 (3.63–9.16) *	22.70 (7.23–71.23) *
*How the parent sees her/his means after the pandemic (ref: better/unchanged)*		
More difficult	2.17 (1.52–3.09) *	1.05 (0.47–2.35)
Much more difficult/seriously at risk	3.74 (2.32–6.01) *	2.08 (0.82–5.24)
*Any mood swing in your children? (ref: no)*		
Yes, more nervous, troubled or sad	1.23 (0.89–1.70)	2.43 (1.07–5.53) *
Yes, their mood has improved	0.87 (0.33–2.30)	1.36 (0.17–11.03)

* Significant at the 1% level (*p* < 0.01). Note: The following variables were discarded at bivariate analyses: age of the youngest or only child (*p* = 0.889), children’s disorders and disabilities (*p* = 0.280), family composition (*p* = 0.412), other relatives at home (*p* = 0.822), relatives infected with SARS-CoV-2 (*p* = 0.687), job loss (*p* = 0.290), activities missed by children (*p* = 0.386), and feeling of loneliness (*p* = 0.489). The *p*-value shown in parentheses is the lowest obtained from the two logistic regression models by means of the Wald test.

**Table 4 nutrients-13-00121-t004:** Multivariable logistic regression analysis of increased and decreased vs. unchanged children’s food intake after COVID-19 outbreak. Results are presented as odds ratios (ORs) and 99% confidence intervals (CIs).

Characteristic	Increased Food Intake	Decreased Food Intake
	OR (99% CI)	OR (99% CI)
*Age, y (ref: ≤30)*		
31–35	0.80 (0.50–1.28)	0.95 (0.54–1.64)
36–40	0.86 (0.55–1.35)	1.03 (0.60–1.75)
41–45	0.79 (0.50–1.26)	0.91 (0.52–1.59)
46–50	0.67 (0.41–1.10)	0.95 (0.51–1.79)
>50	0.60 (0.34–1.07)	0.97 (0.47–2.04)
*Sex (ref: male)*		
Female	1.84 (1.29–2.63) *	1.36 (0.90–2.07)
*Educational attainment of the parents (ref: both secondary school)*		
Secondary school & graduate school	0.95 (0.77–1.17)	0.99 (0.75–1.30)
Both graduate school	0.71 (0.57–0.89) *	0.95 (0.71–1.27)
*Economic status (ref: well-off)*		
Somewhat difficult but overall satisfactory	1.12 (0.92–1.36)	1.25 (0.97–1.61)
Difficult/unsustainable	1.53 (1.03–2.26) *	1.39 (0.82–2.34)
*Number of children (ref: 1)*		
2	1.33 (1.09–1.62) *	1.28 (1.00–1.65)
3	1.79 (1.28–2.51) *	1.34 (0.84–2.15)
>3	1.47 (0.82–2.63)	1.07 (0.48–2.37)
*Age of the youngest or only child, y (ref: ≤2)*		
3–5	1.54 (1.20–1.98) *	0.82 (0.61–1.10)
6–10	1.51 (1.12–2.04) *	0.58 (0.38–0.86) *
11–14	1.91 (1.30–2.80) *	0.87 (0.52–1.48)
>14	1.47 (0.92–2.35)	1.22 (0.67–2.23)
*Economic status after the outbreak (ref: improved/unchanged)*		
Slightly worsened	1.20 (0.98–1.46)	0.96 (0.74–1.24)
Worsened	1.09 (0.70–1.70)	1.02 (0.58–1.79)
*How the parent sees her/his means after the pandemic (ref: better/unchanged)*		
More difficult	1.21 (0.99–1.47)	1.27 (0.98–1.63)
Much more difficult/seriously at risk	1.38 (0.94–2.04)	1.99 (1.24–3.18) *
*Increased worry about running out of food (ref: no)*		
Yes	1.27 (0.95–1.70)	1.15 (0.78–1.69)
*Children have missed outdoor activities (ref: no)*		
Yes	1.23 (1.03–1.46) *	1.07 (0.86–1.35)
*Children have missed playing sports (ref: no)*		
Yes	1.15 (0.97–1.37)	1.03 (0.82–1.29)
*Any mood swing in your children? (ref: no)*		
Yes, more nervous, troubled or sad	2.57 (2.02–3.27) *	2.51 (1.82–3.47) *
Yes, their mood has improved	3.28 (1.93–5.55) *	2.13 (0.96–4.69)
*Did your children have feelings of loneliness? (ref: no)*		
Yes, not putting it into words	1.39 (1.11–1.76) *	1.77 (1.31–2.39) *
Yes, putting it into words	1.68 (1.34–2.10) *	1.97 (1.46–2.66) *

* Significant at the 1% level (*p* < 0.01). Note: The following variables were discarded at bivariate analyses: country of origin (*p* = 0.341), area of residence (*p* = 0.305), working condition (*p* = 0.507), job type (*p* = 0.556), children’s disorders and disabilities (*p* = 0.997), family composition (*p* = 0.409), other relatives at home (*p* = 0.348), type of accommodation (*p* = 0.701), relatives infected with SARS-CoV-2 (*p* = 0.614), smart working (*p* = 0.779), furlough (*p* = 0.327), job loss (*p* = 0.402), and food shortage (*p* = 0.254). The *p*-value shown in parentheses is the lowest obtained from the two logistic regression models by means of the Wald test.

**Table 5 nutrients-13-00121-t005:** Multivariable logistic regression analysis of increased and decreased vs. unchanged children’s body weight after COVID-19 outbreak. Results are presented as odds ratios (ORs) and 99% confidence intervals (CIs).

Characteristic	Weight Gain	Weight Loss
	OR (99% CI)	OR (99% CI)
*Age, y (ref: ≤30)*		
31–35	0.66 (0.40–1.09)	0.98 (0.38–2.55)
36–40	0.86 (0.54–1.38)	1.29 (0.52–3.18)
41–45	0.95 (0.59–1.53)	1.35 (0.53–3.45)
46–50	0.97 (0.58–1.63)	1.88 (0.70–5.04)
>50	1.19 (0.66–2.16)	1.48 (0.49–4.42)
*Sex (ref: male)*		
Female	1.35 (0.97–1.87)	1.34 (0.71–2.55)
*Economic status (ref: well-off)*		
Somewhat difficult but overall satisfactory	1.09 (0.91–1.31)	1.51 (1.05–2.16) *
Difficult/unsustainable	1.24 (0.85–1.81)	1.98 (1.01–3.86) *
*Number of children (ref: 1)*		
2	1.45 (1.18–1.77) *	0.98 (0.68–1.41)
3	1.86 (1.31–2.64) *	1.67 (0.88–3.19)
>3	1.79 (0.99–3.21)	1.49 (0.49–4.49)
*Age of the youngest or only child, y (ref: ≤2)*		
3–5	1.15 (0.89–1.48)	0.75 (0.45–1.26)
6–10	1.41 (1.04–1.91) *	1.28 (0.70–2.33)
11–14	1.25 (0.84–1.85)	1.55 (0.74–3.26)
>14	0.86 (0.53–1.38)	3.19 (1.44–7.09) *
*Children with disorders or disabilities (ref: no)*		
Learning disabilities	1.39 (0.92–2.10)	2.21 (1.09–4.46) *
Other conditions	1.11 (0.73–1.68)	1.54 (0.75–3.18)
Unspecified	1.02 (0.62–1.69)	1.46 (0.61–3.51)
*Increased worry about running out of food (ref: no)*		
Yes	0.84 (0.62–1.14)	0.78 (0.44–1.38)
*Changes in children’s food intake (ref: no)*		
More food	7.25 (6.02–8.74) *	0.72 (0.36–1.44)
Less food	0.60 (0.42–0.87) *	14.43 (10.10–20.6) *
*Children have missed going to school (ref: no)*		
Yes	1.19 (0.99–1.44)	0.82 (0.57–1.18)
*Did your children have feelings of loneliness? (ref: no)*		
Yes, not putting it into words	1.24 (0.99–1.55)	1.20 (0.77–1.87)
Yes, putting it into words	1.43 (1.15–1.77) *	1.70 (1.11–2.61) *

* Significant at the 1% level (*p* < 0.01). Note: The following variables were discarded at bivariate analyses: country of origin (*p* = 0.858), area of residence (*p* = 0.286), educational attainment (*p* = 0.412), working condition (*p* = 0.324), job type (*p* = 0.780), family composition (*p* = 0.893), other relatives at home (*p* = 0.450), type of accommodation (*p* = 0.918), relatives infected with SARS-CoV-2 (*p* = 0.264), smart working (*p* = 0.446), furlough (*p* = 0.790), economic status after the outbreak (*p* = 0.256), job loss (*p* = 0.369), expected means (*p* = 0.292), and food shortage (*p* = 0.444). The *P*-value shown in parentheses is the lowest obtained from the two logistic regression models by means of the Wald test.

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
