# Peer review of "Parents’ Perception of Food Insecurity and of Its Effects on Their Children in Italy Six Months after the COVID-19 Pandemic Outbreak"

_nutrients, 2020, doi:10.3390/nu13010121_

Round 1
Reviewer 1 Report
The title does not seem to be relevant and it can be revised. One suggestion, “Parents’ perception of children’s food intake and food insecurity in Italy six months after the COVID-19 pandemic outbreak” 

Please revise the manuscript to increase its clarity and accuracy.
P1, line 20: what does "a solid household economy" mean?
P1, lines 25-27: Implications for system level can be to provide assistance to the groups in needs or promoting healthy lifestyle within the scope of the pandemic instead of ad hoc measures for fragile groups such as children.
P2, lines 62-64: Revise the objective because parent’s perceptions of primary measures were assessed instead of actually measuring them.
P2, lines 66 - 78: Please add a brief description of “Hunger Vital Sign” including reliability and validity data since food insecurity is the primary measure in this study.
P3, line 85: “in our hospital” can be changed to “in a hospital” and then add a brief description of the hospital. 

P2, line 86: “snowball technique” —> snowball sampling technique
P3, line 92: Add a significance level used for data analysis
P3, line 98: “multivariable regression” —> Multivariable logistic regression 

P7, line 146: “an increase in FI” —> becoming food insecure

P11, line 187-189: Delete this paragraph.
P12, lines 221-223: The option shouldn’t be included without supporting evidence or data
P13, lines 247-248: “we found a strong correlation between children’s weight loss and age older than 14 years” - wording should be improved. 

P13, lines 267-276: Add additional limitations, actual changes in variables were not measured but perceived changes were assessed, for example, actual food consumption was not measured but perceived changes in food intakes were assessed. In addition, the perception of economic state was measured instead of actual economic status.
P13, lines 278-279: “In conclusion, our study suggests that children and their families are already suffering from an increase in FI even in the wealthier Italians areas” -The prevalence of FI was increased but this conclusion statement seems to be overstated.
P13, line 280: What does “disposable means” mean?
P13, line 281: “will” —> may or can
P13, lines 285-286: “Science should work to provide useful information to fill in the major knowledge gaps”: Add more specific info on what kind of knowledge gap should be measured? 

Figures 1 and 2: Frequency —> Percent
Table 4: “more food” —> increased food intake; “less food” —> decreased food intake
Author Response
The title does not seem to be relevant and it can be revised. One suggestion, “Parents’ perception of children’s food intake and food insecurity in Italy six months after the COVID-19 pandemic outbreak”
- Thank you. The title was modified as follows “Parents’ perception of food insecurity and of its effects on their children in Italy six months after the COVID-19 pandemic outbreak”.
Please revise the manuscript to increase its clarity and accuracy.
P1, line 20: what does "a solid household economy" mean?
- “solid” was changed into “wealthy” to better explain the meaning (line 21)
P1, lines 25-27: Implications for system level can be to provide assistance to the groups in needs or promoting healthy lifestyle within the scope of the pandemic instead of ad hoc measures for fragile groups such as children.
- Thank you for your suggestion. The conclusion of the abstract was modified accordingly: “…our findings might serve as a warning to politicians to promote healthy lifestyles and provide assistance to the groups in need.”. (lines 27-28)
P2, lines 62-64: Revise the objective because parent’s perceptions of primary measures were assessed instead of actually measuring them.
- Thank you for your suggestion. According to your and another reviewer’s opinions, the text was modified as follows: “Following these considerations, we hypothesized that social distancing measures to fight COVID-19 pandemic would strongly impact on a vulnerable population such as children in terms of FI and related inadequate nutrition issues. To investigate this hypothesis, we inquired about Italian parents’ perception of the effects of the lockdown imposed for SARS-CoV-2 infection containment on FI, eating habits, and weight changes on their children 6 months after the beginning of the pandemic.” (lines 64-68)
P2, lines 66 - 78: Please add a brief description of “Hunger Vital Sign” including reliability and validity data since food insecurity is the primary measure in this study.
- Thank you for your request. In order not to make the “Methods” section too busy, we included a description of the HVS including validation data among different age groups in Appendix A. This is the text that was added: “The first item reports “Within the past 12 months we worried whether our food would run out before we got money to buy more”, and the second “Within the past 12 months the food we bought just didn’t last and we didn’t have money to get more”; affirmative answers are “often true” or “sometimes true”, whereas “never true” indicates no FI. The HVS has been validated among young children (97% sensitivity, 83% specificity) [27], adolescents and youth (88.5% sensitivity, 84.1% specificity) [77], and in adults (>97% sensitivity, >70% specificity, with higher values among high-risk subgroups) [78]. The HVS, as a major limit, does not allow to assess the severity of food insecurity.” (lines 401-408)
P3, line 85: “in our hospital” can be changed to “in a hospital” and then add a brief description of the hospital.
- The phrase was changed into “…in a tertiary care pediatric hospital…” (line 90)
P2, line 86: “snowball technique” —> snowball sampling technique
- “sampling” was added. Thank you. (line 91)
P3, line 92: Add a significance level used for data analysis
Reply: Significance level specification can be found at the very end of the Methods section. It has been lowered from 0.05 to 0.01 following the suggestion of another reviewer. (lines 115-117)
P3, line 98: “multivariable regression” —> Multivariable logistic regression
- Thank you, “logistic” was added. (line 105)
P7, line 146: “an increase in FI” —> becoming food insecure
- Thank you, the sentence was modified according to your suggestion. (line 165)
P11, line 187-189: Delete this paragraph.
- Thank you for spotting this inaccuracy. These lines were removed.
P12, lines 221-223: The option shouldn’t be included without supporting evidence or data
- Your consideration is correct, the sentence was removed.
P13, lines 247-248: “we found a strong correlation between children’s weight loss and age older than 14 years” - wording should be improved.
- Thank you. The sentence was rephrased as follows “Interestingly, we found that children older than 14 years were significantly at risk of weight loss during lockdown.”. (lines 306-307)
P13, lines 267-276: Add additional limitations, actual changes in variables were not measured but perceived changes were assessed, for example, actual food consumption was not measured but perceived changes in food intakes were assessed. In addition, the perception of economic state was measured instead of actual economic status.
- We thank the reviewer for this comment. We added an additional sentence to the limitations section: “Similarly, the questionnaire did not measure the actual food consumption, but the perceived changes in food intake, nor the actual economic status, but its perception.”. (lines 345-347)
P13, lines 278-279: “In conclusion, our study suggests that children and their families are already suffering from an increase in FI even in the wealthier Italians areas” -The prevalence of FI was increased but this conclusion statement seems to be overstated.
- The sentence was softened by changing “are” into “may be”: “children and their families may be already suffering from an increase in FI even in the wealthier Italian areas” (line 354)
P13, line 280: What does “disposable means” mean?
- The expression was unclear, and it was removed.
P13, line 281: “will” —> may or can
- Thank you. “Will” was changed into “may”. (line 357)
P13, lines 285-286: “Science should work to provide useful information to fill in the major knowledge gaps”: Add more specific info on what kind of knowledge gap should be measured?
- The sentence was modified as follows: “Scientific research should work to fill in the major knowledge gaps about food and nutrition implications of the pandemics and effective delivery of equitable social protection programmes and policies in these circumstances.[31]” (lines 361-363)
Figures 1 and 2: Frequency —> Percent
- We decided to use “%” as the title of the Y-axis of all charts because figure captions are quite long. We think that adding “percent” would make the captions less clear.
Table 4: “more food” —> increased food intake; “less food” —> decreased food intake
- Thank you. This was changed according to your suggestion.
Reviewer 2 Report
Dear Authors,
The general idea of ​​the study is interesting, but the description should be strongly improved.
Introduction
- If we considers the food insecurity in the context of malnutrition and overnutrition, attention should be paid to the problem of hidden hunger, which may arise in conditions of limited access to food. You should mention that.
- At the end of this section, I suggest that the Authors formulate research hypotheses, which is usually practiced..
Material and methods
- It is necessary to present a research scheme (sample collection chart) with successive stages of the procedure and excluding incomplete cases.
- What test was used to assess the significance of OR ?- please explain.
Results
In my opinion, it is a mistake to miss individual responses for a given variable (for age - 1 missing, for country of origin -29 missing, etc.). All cases containing any gaps should be eliminated in the entire survey (6094 respondents). In addition, I believe that this type of study can ignore the group of men, which is only 8% of the respondents. The inclusion of men in the logistic regression analysis could have caused confusion in the final results (Tables 3 and 4). I suggest doing the analysis for women only and comparing the results, because they will probably be different, and this affects the conclusions.
Discussion
- Lines 187-189 come from the instructions for Authors - they should be removed.
- I suggest you comment on the information on line 206.
- Lines 224-236. Here, the authors confirm the similarity of the obtained results to others, and it is necessary to comment on the probable cause of the phenomenon and its implications.
- Lines 247-251. I suggest you discuss this information in a broader context and propose possible age-related causes of weight loss.
- At the end of the discussion, it would be good to provide the direction of verification of the research hypothesis and briefly summarize the results of the research.
Appendix A.
Lines 307-312 should be removed because they are the content of the instructions for Authors.
Regards
Author Response
Dear Authors,
The general idea of ​​the study is interesting, but the description should be strongly improved.
Introduction
- If we considers the food insecurity in the context of malnutrition and overnutrition, attention should be paid to the problem of hidden hunger, which may arise in conditions of limited access to food. You should mention that.
- Thank you for your suggestion. We have now mentioned “hidden hunger” in the introduction as follows: “...through inadequate nutrition in terms of hunger, undernutrition, overnutrition with low-quality, sugary and fatty food, and an insufficient intake of macronutrients and micronutrients, often referred to as “hidden hunger” to describe the invisible nature of the problem and the lack of overt symptoms.” (lines 37-38)
- At the end of this section, I suggest that the Authors formulate research hypotheses, which is usually practiced.
- The research hypotheses were added at the end of the introduction section as follows: “Following these considerations, we hypothesized that social distancing measures to fight COVID-19 pandemic would strongly impact on a vulnerable population such as children in terms of FI and related inadequate nutrition issues. To investigate this hypothesis, we inquired about Italian parents’ perception of the effects of the lockdown imposed for SARS-CoV-2 infection containment on FI, eating habits, and weight changes on their children 6 months after the beginning of the pandemic.” (lines 64-68)
Material and methods
- It is necessary to present a research scheme (sample collection chart) with successive stages of the procedure and excluding incomplete cases.
Reply: A flow chart showing the exclusion of incomplete cases has been included in Appendix B (Figure A1).
- What test was used to assess the significance of OR ?- please explain.
Reply: Thank you for your suggestion. We have added this piece of information at the very end of the Methods section.(lines 116-117)
Results
In my opinion, it is a mistake to miss individual responses for a given variable (for age - 1 missing, for country of origin -29 missing, etc.). All cases containing any gaps should be eliminated in the entire survey (6094 respondents). In addition, I believe that this type of study can ignore the group of men, which is only 8% of the respondents. The inclusion of men in the logistic regression analysis could have caused confusion in the final results (Tables 3 and 4). I suggest doing the analysis for women only and comparing the results, because they will probably be different, and this affects the conclusions.
Reply: All the analyses have been rerun on the subsample of 5811 records with complete data, as suggested. Participants with no data on smart working and/or children’s disabilities have been retained in the analysis due to the large portion of missing data on these two variables and treated as a different category or imputed with recursive procedures, respectively. This has now been clarified in the Methods and Results sections. With reference to your second point, we believe that confusion can arise only if a variable that is treated as a potential confounder is actually a strong effect modifier that changes the association between exposure and outcome. We tested for the presence of significant interactions in the model for food insecurity (FI) and, of note, we found that the strongest one was that between sex and educational attainment (P=0.044), suggesting that highest education might be a stronger protective factor against FI in females than in males. We think that this evidence does not justify the execution of stratified analyses or the removal of ≈500 subjects from the study sample.
Discussion
- Lines 187-189 come from the instructions for Authors - they should be removed.
- Thank you for spotting this inaccuracy. These lines were removed.
- I suggest you comment on the information on line 206.
- Thank you for your request. We added the following sentence: “The latter are likely people who were already subject to an unstable household economy, and in whom the containment measures for the pandemic caused a dramatic worsening of their already precarious condition.”. (lines 238-240)
- Lines 224-236. Here, the authors confirm the similarity of the obtained results to others, and it is necessary to comment on the probable cause of the phenomenon and its implications.
- Thank you for your comment. We added the following sentences, and we think that raising these points adds much to the discussion.
“The risk for FI drops quickly with income [41], and several authors report that also households with children, higher size households, a single, divorced or separated parent, especially if a woman, or an unmarried parent in a more complex family (for example, one that includes a cohabiting partner or another adult such as a grandparent) are factors more frequently associated with FI [42-45]. Furthermore, a renter, a younger, or a less educated person is more likely to be food insecure than their respective counterparts [46].” (lines 248-254)
“As a matter of fact, global Covid-19 data reveal a concrete and urgent threat to food security [49,50]. The number of people suffering from chronic hunger can rise dramatically, and the situation may worsen for those who already suffer from FI [51]. The reduction of access to food and other essential items is expected to magnify disparities in healthy lifestyle behaviors and increase social tensions, migration, and severe malnutrition, which has been associated with a higher risk for more severe cases of SARS-CoV-2 infection [52]. International agencies point out the necessity to adopt measures to facilitate the flow of food, support the most vulnerable, ensure access to adequate and healthy food, and invest in sustainable and resilient food systems [53].” (lines 262-269)
- Lines 247-251. I suggest you discuss this information in a broader context and propose possible age-related causes of weight loss.
- Thank you for your request. We broadened the discussion about weight loss among adolescents and added several references about this topic. “Among teenagers, evidence supports a strong association between weight/shape concern and low self-esteem, as well as emotional problems, including depressive symptoms and anxiety [67]. In a school-based survey on 4746 adolescents, the relationship between body dissatisfaction and self-esteem was strong and significant in both sexes, and did not differ significantly between middle school and high school cohorts, indicating that body dissatisfaction and self-esteem are strongly related among nearly all groups of adolescents [69]. Both low self-esteem and body dissatisfaction early in life have been found to predict adverse health outcomes later in life, including the use of unhealthy weight-control behaviors, other eating-disordered behaviors, general psychological distress, and a variety of other negative outcomes [70-72]. In stressful times like those of an active pandemic with severe containment measures, monitoring such possible effects among teenagers should not be forgotten.” (lines 310-320)
- At the end of the discussion, it would be good to provide the direction of verification of the research hypothesis and briefly summarize the results of the research.
- We added a short paragraph including these requests at the end of the discussion: “Despite these limitations, this study seems to confirm our initial hypothesis that social distancing measures to fight COVID-19 pandemic may be impacting on vulnerable populations such as children and their families in terms of worsening in FI and related inadequate nutrition issues with consequent weight changes which can be worrisome in specific age groups.” (lines 348-351)
Appendix A.
Lines 307-312 should be removed because they are the content of the instructions for Authors.
- Thank you for spotting this inaccuracy. These lines were removed.
Reviewer 3 Report
This is an interesting study which examines attitudes and experiences to COVID-19, food insecurity, food intake, and body weight. The are numerous minor edits to the writing which are needed (in particular, "CODIV").
My main concern is that a large sample size was used, there are many tests, and the significance of 0.05 is used. A correction for the number of tests used is needed. It was unclear exactly how many variables were included in the model, before the "non-significant" results were excluded
In the introduction this paper states that FI and eating habits are examined in regards to lockdown (lines 62-64). The inclusion of body weights in the result was a surprise.
The authors discuss that covariates that were not significant (at P<0.25) at the bivariate level were excluded from the regression (line 98). I cannot see the bivariate results reported, for instance, was a member of the family getting COVID-19 not associated with FI, if not, what was the P.value?
It is unclear how some of the variables were obtained and calculated. I did not find Appendix A overly helpful in understanding the variables. For instance, what was the exact question asked regarding economic status after the outbreak? Including enough detail so that other studies can reproduce this study are important.
Can the rationale for some of the covariates be explained in more detail. For instance, was there an anticipated association between children missing outdoor activities and more/less food intake.
Other comments:
line 47: suggest state February 24 2020 (include year)
The tables are difficult to read as the characteristics and the characteristic levels and not easily distinguishable. Can the characteristics be bolded, italicized, or differentiated somehow to make for easier reading.
Discussion and Appendix A, presumably text from the Nutrients guidelines to authors have been included here.
Author Response
This is an interesting study which examines attitudes and experiences to COVID-19, food insecurity, food intake, and body weight. The are numerous minor edits to the writing which are needed (in particular, "CODIV").
- Thank you for your kind appreciation. “CODIV” had been actually written in the table captions and corrected throughout.
My main concern is that a large sample size was used, there are many tests, and the significance of 0.05 is used. A correction for the number of tests used is needed. It was unclear exactly how many variables were included in the model, before the "non-significant" results were excluded
Reply: Thank you for your suggestion. The significance level was lowered from 0.01 to 0.05 in order to control for type I error related to multiple testing. Other and more specific correction methods such as Bonferroni’s or Šidák’s have no direct application in multivariable regression analysis.
In the introduction this paper states that FI and eating habits are examined in regards to lockdown (lines 62-64). The inclusion of body weights in the result was a surprise.
- Following your suggestion, we added “weight changes” to the end of the introduction as follows: “...we inquired about Italian parents’ perception of the effects of the lockdown imposed for SARS-CoV-2 infection containment on FI, eating habits, and weight changes on their children 6 months after the beginning of the pandemic.” (lines 66-68)
The authors discuss that covariates that were not significant (at P<0.25) at the bivariate level were excluded from the regression (line 98). I cannot see the bivariate results reported, for instance, was a member of the family getting COVID-19 not associated with FI, if not, what was the P.value?
Reply: Thank you for raising this point. Results of bivariate analyses are now reported in the footnotes of Tables 3–5.
It is unclear how some of the variables were obtained and calculated. I did not find Appendix A overly helpful in understanding the variables. For instance, what was the exact question asked regarding economic status after the outbreak? Including enough detail so that other studies can reproduce this study are important.
Reply: Thank you. A detailed description of the questions regarding the economic status was included in Appendix A. (lines 391-398) More information about the Hunger Vital Sign was also added in Appendix A, as requested by another reviewer. (lines 401-408)
Can the rationale for some of the covariates be explained in more detail. For instance, was there an anticipated association between children missing outdoor activities and more/less food intake.
R.: Your point allowed us to improve the discussion by adding some considerations on some of the variables that were analyzed (missing outdoor activities and food intake, poverty, and weight gain/loss and feelings of loneliness), so as to be even more exhaustive. (lines 282-286, 296-298, 299-305)
Other comments:
line 47: suggest state February 24 2020 (include year)
- “2020” was added to the date as suggested.
The tables are difficult to read as the characteristics and the characteristic levels and not easily distinguishable. Can the characteristics be bolded, italicized, or differentiated somehow to make for easier reading.
Reply: We have italicized variable names in all tables, as suggested
Discussion and Appendix A, presumably text from the Nutrients guidelines to authors have been included here.
- Thank you for spotting these inaccuracies. These paragraphs were removed.
Round 2
Reviewer 2 Report
Dear Authors,
thank you for considering my suggestions. At the same time, I accept your explanations.
Sincerely Yours